

# Crystal structure of human Acinus RNA recognition motif domain

Humberto Fernandes[1,2,*], Honorata Czapinska[1,*], Katarzyna Grudziaz[1], Janusz M. Bujnicki[1,3] and Martyna Nowacka[1,4]

[1] International Institute of Molecular and Cell Biology in Warsaw, Warsaw, Poland
[2] Institute of Biochemistry and Biophysics, Polish Academy of Sciences, Warsaw, Poland
[3] Institute of Molecular Biology and Biotechnology, Faculty of Biology, Adam Mickiewicz University, Poznan, Poland
[4] Current Affiliation: University of Warsaw Biological and Chemical Research Centre, Warsaw, Poland
* These authors contributed equally to this work.

## ABSTRACT

Acinus is an abundant nuclear protein involved in apoptosis and splicing. It has been implicated in inducing apoptotic chromatin condensation and DNA fragmentation during programmed cell death. Acinus undergoes activation by proteolytic cleavage that produces a truncated p17 form that comprises only the RNA recognition motif (RRM) domain. We have determined the crystal structure of the human Acinus RRM domain (AcRRM) at 1.65 Å resolution. It shows a classical four-stranded antiparallel β-sheet fold with two flanking α-helices and an additional, non-classical α-helix at the C-terminus, which harbors the caspase-3 target sequence that is cleaved during Acinus activation. In the structure, the C-terminal α-helix partially occludes the potential ligand binding surface of the β-sheet and hypothetically shields it from non-sequence specific interactions with RNA. Based on the comparison with other RRM-RNA complex structures, it is likely that the C-terminal α-helix changes its conformation with respect to the RRM core in order to enable RNA binding by Acinus.

## INTRODUCTION

The RNA recognition motif (RRM) is a small (approximately 90 residues) protein domain known for its characteristic βαββαβ fold and frequent engagement in RNA binding. RRM-containing proteins are typically engaged in splicing, editing, export, degradation and regulation of translation (*Maris, Dominguez & Allain, 2005*). A single RRM typically recognizes a short continuous stretch of single-stranded RNA, usually less than eight ribonucleotides long (*Afroz et al., 2015*). Binding involves two conserved motifs termed RNP1 and RNP2, located in the two central β-strands, with [RK]-G-[FY]-[GA]-[FY]-[ILV]-X-[FY] and [ILV]-[FY]-[ILV]-X-N-L consensus sequences, respectively (X stands for any amino acid) (*Maris, Dominguez & Allain, 2005*; *Clery, Blatter & Allain, 2008*). Prominent features of these motifs include conserved aromatic residues that are often involved in interactions with ribonucleotides of the RNA target sequence. In addition to

Corresponding authors
Janusz M. Bujnicki,
iamb@genesilico.pl
Martyna Nowacka,
mnowacka@genesilico.pl

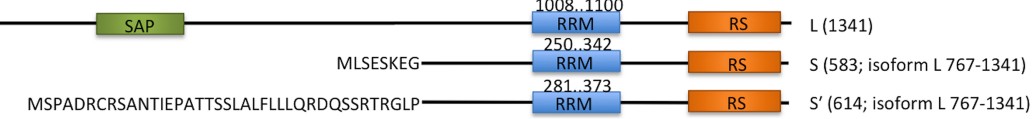

**Figure 1** Schematic overview of three endogenous Acinus isoforms (L, S and S′) with SAP domain in green, RNA recognition motif (RRM) in blue, and RS-like domain in orange. The different N-termini of Acinus S and S′ isoforms are indicated by amino acid sequence. The length of each isoform is given in brackets. The position of RRM domain in each isoform is given above the blue rectangles.

the RRM core architecture, other structural elements may or may not be present: β-hairpins (*Volpon et al., 2005*), extra β-strands (*Oberstrass et al., 2005*; *Blatter et al., 2015*) or α-helices (*Jacks et al., 2003*). These additional elements, together with variable loops, can contribute to the recognition of an RNA target.

Acinus (apoptotic chromatin condensation inducer in the nucleus) is a protein involved in apoptosis and alternative splicing regulation. Human Acinus has three isoforms, Acinus-L, Acinus-S and Acinus-S′, which are most probably generated by alternative splicing and/or usage of alternative promoters (*Sahara et al., 1999*). The isoforms differ in size and N-termini (Fig. 1). The longest isoform L has a putative DNA binding domain, SAP, and two shorter isoforms, S and S′, have unique sequences at their N-termini (*Sahara et al., 1999*). All of them contain an RRM domain at the C-terminus and a conserved arginine/serine-repeat (RS repeat) region that are frequently present in splicing factors.

It was originally discovered that Acinus is activated by caspase-3, which recognizes its consensus DELD target sequence and cleaves it after D1093 (residue numbering of the Acinus-L isoform is assumed throughout the text). The truncated version of Acinus (called p17, residues 987–1093) covers the RRM domain and induces chromatin condensation at the terminal stages of apoptosis (*Sahara et al., 1999*; *Hu et al., 2005*). The cleavage of Acinus by caspase-3 can be inhibited by phosphorylation of S1180 and S1331 (*Hu et al., 2005*) or by the antiapoptotic AAC-11 (antiapoptosis clone 11) protein which directly binds to Acinus in the region comprising residues 840–918 (*Rigou et al., 2009*).

In other studies, knock downs of Acinus did not prevent apoptotic chromatin condensation but instead impaired DNA fragmentation, suggesting involvement in DNA cleavage during programmed cell death (*Joselin, Schulze-Osthoff & Schwerk, 2006*; *Rigou et al., 2009*). Later, Acinus was also found in ribonucleoprotein complexes: the spliceosome (*Rappsilber et al., 2002*; *Zhou et al., 2002*), the apoptosis- and splicing-associated protein (ASAP) complex (*Schwerk et al., 2003*; *Murachelli et al., 2012*) and the splicing-dependent multiprotein exon junction complex (EJC) (*Tange et al., 2005*). Consequently, its function in splicing became apparent. Acinus was shown to regulate splicing of a subset of apoptotic genes in human cells (*Michelle et al., 2012*) and to facilitate constitutive splicing of pre-mRNAs containing a weak, alternative 5′ splice site, favoring the mRNA isoform generated from this site (*Rodor et al., 2016*). Other results coupled Acinus to transcription and splicing of retinoic acid receptor (RAR)-dependent genes (*Vucetic et al., 2008*; *Wang, Soprano & Soprano, 2015*). Moreover, the RRM domain was

found necessary for the enhancement of Acinus activity on the retinoic acid (RA) dependent splicing of a mini-gene from its weak 5′ splice site (*Wang, Soprano & Soprano, 2015*).

The RNA binding mode of Acinus remains unknown. Recently, however, Acinus-S was found to bind to a subset of suboptimal introns of pre-mRNAs, as well as to spliced mRNAs (*Rodor et al., 2016*). The analysis of iCLIP data (individual-nucleotide resolution UV crosslinking and immunoprecipitation) for Acinus-S demonstrated an enrichment of two sequence motifs: a U-rich sequence (in introns), and a "GAAGAA"-like motif (in exons). Additionally, it was shown that Acinus regulates the splicing of DFFA/ICAD that is an inhibitor of caspase-activated DNase and a major regulator of DNA fragmentation (*Rodor et al., 2016*).

Here, we report for the first time a crystal structure of the human Acinus RRM domain (AcRRM) at 1.65 Å resolution together with the analysis of its putative RNA binding mode. The structure of an AcRRM monomer shows a classic four-stranded β-sheet fold with two flanking α-helices and an additional non-classical α-helix at the C-terminus. In the absence of the target RNA, the C-terminal α-helix partially occludes the potential ligand binding site (RNP1 and RNP2).

## MATERIALS AND METHODS

### Cloning, expression and purification of Acinus RRM

The protein fragment encoding the AcRRM domain was selected based on the secondary structure prediction of Acinus protein performed with the GeneSilico Metaserver (*Kurowski & Bujnicki, 2003*). The human Acinus cDNA (ID 9021673) was purchased from IMAGE Consortium Source BioScience. The DNA sequence encoding AcRRM (residues 1008–1100, Uniprot sequence ID Q9UKV3) was PCR amplified and subcloned into pGEX-4T1 (GE Healthcare Life Sciences, Little Chalfont, UK). *E. coli* BL21(DE3) strain (New England BioLabs, Ipswich, MA, USA) was used to overexpress the glutathione S-transferase (GST) tagged AcRRM. Expression was carried out in LB medium, induced with 1 mM isopropyl-D-1-thiogalactopyranoside solution (IPTG) at $OD_{600}$ of 0.6 and conducted at 37 °C with shaking at 200 rpm for 4 h. The cells were harvested by centrifugation at 4,000 × g, at 4 °C for 20 min. The pellet was resuspended in 1× PBS buffer, pH 7.9, supplemented with 1 mM phenylmethylsulfonyl fluoride (PMSF) (Sigma-Aldrich, St. Louis, Missouri, USA). The cells were subsequently lysed by the French press at 18,000 psi and the cell debris was removed by centrifugation at 20,000 × g for 30 min. The GST-AcRRM fusion protein was purified by GST affinity with Glutathione-Agarose beads (Sigma-Aldrich, St. Louis, Missouri, USA) according to the manufacturer protocol (Fig. S1A). The affinity-purified recombinant protein contained the thrombin recognition site and was subsequently cleaved with thrombin from bovine plasma (Sigma-Aldrich, St. Louis, Missouri, USA) to remove the GST tag (100 μg lyophilized thrombin was used to cleave 1 mg of GST-AcRRM fusion). The cleavage was performed for 2–4 h at room temperature in buffer containing 50 mM Tris–HCl pH 9, and 50 mM NaCl. After cleavage, AcRRM maintained two additional amino acid residues (GS) at its N-terminus. AcRRM was separated from GST tag by heparin affinity using a HiTrap Heparin HP column (GE Healthcare Life Sciences, Uppsala, Sweden) (Fig. S1B), pre-equilibrated in buffer containing

50 mM Hepes-NaOH pH 8 and 1 mM DTT. The bound protein was eluted with gradient of 0 to 1 M NaCl in the 50 mM Hepes-NaOH pH 8 and 1 mM DTT buffer.

Finally, AcRRM was purified by size exclusion chromatography on a Superdex 75 pg column (GE Healthcare Life Sciences, Little Chalfont, UK) pre-equilibrated in 50 mM Hepes-NaOH pH 8, 100 mM NaCl, 1 mM DTT buffer (Fig. S1C). The purified AcRRM protein was concentrated to 8 mg/ml using a 3,000 molecular-weight cut-off Amicon Ultra-4 Centrifugal Filter Units (Merck Millipore, Burlington, MA, USA).

## Crystallization

Human AcRRM was crystallized at 18 °C using 0.2 + 0.2 µl sitting drop vapor diffusion method. The Phoenix nano-dispensing robot (Art Robbins instruments, Sunnyvale, CA, USA) was used to set the crystallization drops with the Morpheus Screen (Molecular Dimensions), Index and Crystal Screen (Hampton Research, Aliso Viejo, CA, USA) in 96 well crystallization plates (Hampton Research, Aliso Viejo, CA, USA). The first needle-like crystals appeared within two days. The crystals collected from Index Screen 81 (G9) crystallization reagent (0.2 M Ammonium acetate, 100 mM Tris–HCl pH 8.5, 25% w/v Polyethylene glycol 3,350) were cryo-protected for 10 s in reservoir solution supplemented with 25% w/v Polyethylene glycol 400, flash-frozen, stored in liquid nitrogen and used for X-ray data collection.

## X-ray data collection and structure determination

Native X-ray diffraction data sets extending to 1.65 Å resolution were collected using the CCD detector at beamline 14.2 at the Helmholtz-Zentrum Berlin (BESSY II, Germany) (*Gerlach, Mueller & Weiss, 2016*). A data set consisting of 120 frames was collected with an oscillation width of 1° per frame and a crystal-to-detector distance of 150 mm. The data were indexed and scaled using the XDS software package (*Kabsch, 2010*).

Primary molecular replacement searches were performed with the MrBUMP/Phaser pipeline (*McCoy et al., 2007*; *Keegan & Winn, 2008*) using RRM domain of Rna15 (PDB code 2X1F (*Pancevac et al., 2010*)) as a search model. It produced a marginal solution with two monomers in the asymmetric unit, a *Z* score of 11.5 and a final log-likelihood gain (LLG) of 104. The model was then automatically rebuilt with ARP/wARP (*Langer et al., 2008*) excluding some resolution shells that were marked as ice rings. Re-inspection of diffraction images and the Wilson plot revealed two problematic resolution shells (2.29–2.22 and 1.94–1.90 Å). They were then excluded from a second iteration of indexing and scaling. Statistics for the diffraction data collection and (re)processing are presented in Table 1. Iterations of manual rebuilding, using the Coot program (*Emsley & Cowtan, 2004*), and restrained refinement using CCP4 REFMAC5 (*Murshudov et al., 2011*) that included TLS optimization with one group per monomer, produced the AcRRM model characterized by crystallographic *R* factor of 18.5% and $R_{free}$ of 22.4%. The data processing and refinement statistics are summarized in Table 1. Final model coordinates and the corresponding structure factors were deposited at Protein Data Bank with the 6G6S accession code.

**Table 1 Data collection and structure refinement statistics.**

**Data collection**

| | |
|---|---|
| Diffraction source | BESSY 14.2 |
| Wavelength (Å) | 0.918410 |
| Rotation range per image (°) | 1 |
| Total rotation range (°) | 120 |
| Crystal-to-detector distance (mm) | 150 |
| Space group | $P\,2_1\,2_1\,2_1$ |
| Unit cell parameters (Å) | 30.4, 67.9, 80.1 |
| Mosaicity (°) | 0.32 |
| Resolution range (Å) | 80.7–1.65 (1.74–1.65) |
| Excluded resolution ranges (Å) | 2.29–2.22, 1.94–1.90 |
| Total No. of reflections | 90,411 |
| No. of unique reflections | 19,003 |
| Completeness (%) | 91.6 (98.5) |
| Multiplicity | 4.76 (4.80) |
| $<I/\sigma\,(I)>$ [a] | 16.3 (1.97) |
| $R_{meas}$ (%) [b] | 7.1 (86.4) |

**Refinement**

| | |
|---|---|
| Program | REFMAC 5.8.0189 |
| Overall mean B-factor (Å$^2$) for protein chains | 16.0, 15.2 |
| Overall mean B-factor (Å$^2$) for solvent | 30.2 |
| $R_{work}$ (%) [c] | 18.5 |
| $R_{free}$ (%) [d] | 22.4 |
| Rms deviations from ideal | |
| Bond lengths (Å) | 0.017 |
| Bond angles (°) | 1.8 |
| Residues in Ramachandran plot (%) | |
| Favored | 99 |
| Outliers | 0 |

Notes:
  Values in parenthesis are for the outer resolution shell.
  [a] $<I/\sigma\,(I)>$ is the mean signal-to-noise ratio, where $I$ is the integrated intensity of a measured reflection and $\sigma\,(I)$ is the estimated error in the measurement.
  [b] $R_{meas} = 100 \times \Sigma_{hkl}\{N(hkl)/[N(hkl) - 1]\}^{1/2}\,\Sigma_i|I_i(hkl) - <I(hkl)>|/\,\Sigma_{hkl}\Sigma_iI_i(hkl)$, where $I_i(hkl)$ is the $i$th observed intensity of reflection hkl, $<I(hkl)>$ is the average of symmetry-related observations of reflection hkl and $N(hkl)$ is the multiplicity.
  [c] $R_{work} = 100 \times \Sigma_{hkl}\,\|\,F_{obs}| - |F_{calc}\,\|\,/\Sigma_{hkl}|F_{obs}|$, where Fobs and Fcalc are observed and calculated structure-factor amplitudes, respectively.
  [d] The $R_{free}$ value was calculated as for $R_{work}$ using only an unrefined randomly chosen subset of reflection data (5%).

# RESULTS

## Overall structure

Acinus RRM domain exhibits the α/β sandwich fold (Figs. 2A and 2B). The core βαββαβ is followed by an additional C-terminal α-helix not present in the canonical RRM domains. The two molecules in the asymmetric unit of the crystal structure are very similar (rmsd of 0.25 Å for 91 Cα atoms), despite the fact that no non-crystallographic

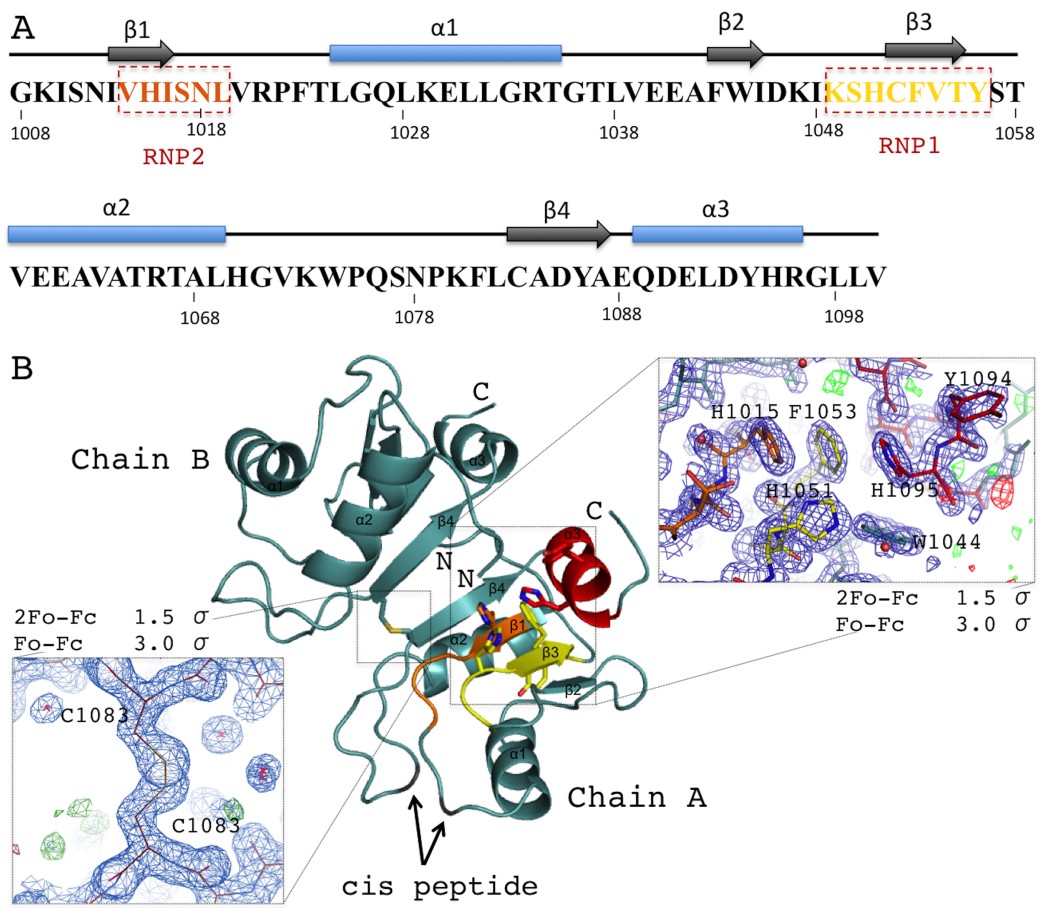

**Figure 2 Overview of the human AcRRM domain (PDB code 6G6S).** (A) Amino-acid sequence of the human AcRRM with the α-helices (blue) and β-strands (gray) annotated above. The RNP2 and RNP1 motifs are indicated by the dashed boxes. (B) The cartoon overview of the secondary structure elements labelled as follows: yellow for RNP1 (on β-strand 3), orange for RNP2 (partially on β-strand 1), red for the C-terminal α-helix; with the electron density views enlarged for the region rich in aromatic residues and the disulfide bond between cysteines 1083 of different chains (most likely a crystal packing artifact). The secondary structure elements are labelled.

symmetry (NCS) restraints were used during the refinement. The N- and C-termini display good electron density for all residues of molecule A, but are not fully ordered in molecule B. AcRRM has three prolines within its sequence and two of them, P1022 and P1075, are in the *cis*-conformation in both molecules. The two AcRRM monomers are covalently bound in the crystal structure via a disulfide bond (Fig. 2B), that generates an interface area of approximately 500 Å for each of them (calculated by the PISA server (*Krissinel, 2015*)). For the two linked AcRRM monomers present in the crystal, the Complexation Significance Score (CSS) calculated by PISA is 1. However, computational modeling of the disulfide bond removal by replacement of C1083 with alanine, results in the CSS of 0. This indicates that in the reducing conditions the dimerization does not take place and the dimeric assembly is likely to be a crystal-packing artifact. The monomeric state of AcRRM in reducing conditions was confirmed by gel

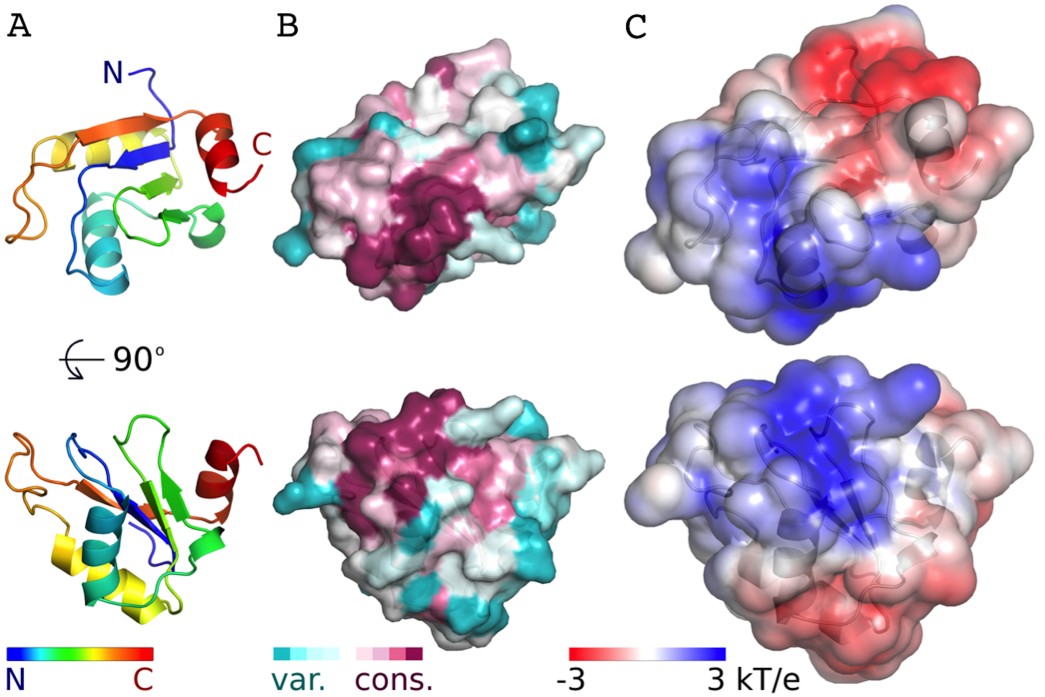

**Figure 3 Conservation and predicted charge distribution mapped on the surface of the AcRRM domain.** (A) Overview of the human AcRRM domain colored according to the position in the sequence (RNP1 and RNP2 are located in the green and blue colored β-strands, the C-terminal α-helix is colored in red). (B) Amino acid sequence conservation of AcRRM calculated using the ConSurf server (*Ashkenazy et al., 2010*) and mapped on the solvent excluded surface of the protein. (C) The electrostatic potential of AcRRM calculated using the DelPhi server (*Li et al., 2012*) and mapped on the solvent accessible surface of the domain. Sequence alignment used for the calculation of ConSurf scores is presented in the Fig. S4.                               

filtration (Fig. S2). C1083 is conserved in AcRRMs of human, mouse and fly, but in other RRM domains this position is generally not conserved (Fig. S3).

RNA recognition motif domains typically bind RNA via the aromatic residues of the RNP1 and RNP2 motifs located in the two central β-strands. These residues are also present in AcRRM. RNP1 comprises residues 1049–1056 [KSHCFVTY], with the deviations found at second, third and fourth position of the consensus sequence (underlined). RNP2 comprises residues 1014–1019 [VHISNL] with deviations at the second position of the consensus (Fig. 2A). The aromatic residues of RNPs are exposed at the β-sheet surface. In their close vicinity, there are additional aromatic amino acids (Y1094 and H1095) of the C-terminal α-helix packed against the β-sheet on the opposite side to the α-helices 1 and 2. Furthermore, the C-terminal α-helix contains the DELD caspase-3 target site.

The AcRRM domain has a single conserved region on the surface, which correlates with a part of the protein that is predicted to be positively charged (Fig. 3, Fig. S4). The region covers the RNP1 and RNP2 strands of the β-sheet and extends towards its edge. It correlates well with the RNA binding sites of the most similar RRM domains. Therefore, it

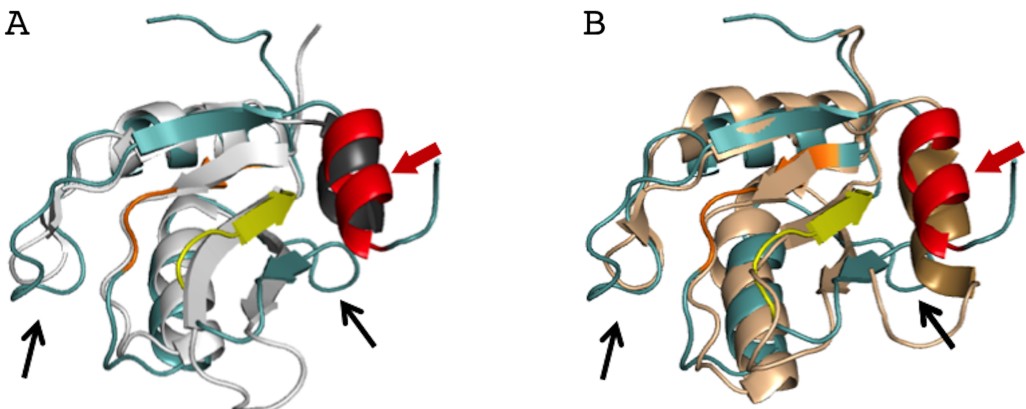

**Figure 4 Superposition of the AcRRM structure with two most similar structures.** (A) RRM of Rna15 (PDB code 2X1B (*Pancevac et al., 2010*)) and (B) RRM of U1A (PDB code 5FJ4 (*Huang, Wang & Lilley, 2016*)). AcRRM is colored as in Fig. 2, and other RRMs are colored in light gray and light brown for panel (A) and (B), respectively, with C-terminal features highlighted in darker shades. The caspase cleavage site in AcRRM is indicated by red arrow, and the two sequence insertions in AcRRM are indicated by black arrows.

seems plausible that Acinus binds its RNA target via this region in a way similar to other RRMs.

## Comparison with other RRMs

The AcRRM crystal structure was compared with other RRM structures deposited in the Protein Data Bank using the DALI server (*Holm & Rosenstrom, 2010*). AcRRM was found to be structurally most similar to RRM of Rna15, a subunit of the 3′-end processing factor from *Saccharomyces cerevisiae* (PDB code: 2X1B (*Pancevac et al., 2010*)). The βαββαβ cores of both structures align with an rmsd of 1.6 Å (sequence identity of 17%) (Fig. 4A). The RRM of U1A (U1 small nuclear ribonucleoprotein A) was found to be the second closest structural relative of AcRRM (PDB: 5FJ4 (*Huang, Wang & Lilley, 2016*)). It harbors the C-terminal α-helix and aligns with an rmsd of 2.1 Å (sequence identity of 17%) (Fig. 4B).

There are two sequence insertions in AcRRM compared to the most similar known RRM structures (Figs. 4 and 5A): the first insertion (residues 1040 and 1041) distorts the following β-strand making it shorter and increases the size of the loop-2 preceding β2. The second insertion (residues 1076 to 1078) is located in loop-5, making it larger than the corresponding loop in both other structures (RRMs of Rna15 and U1A) (Figs. 4 and 5A). In contrast, loop-3 is substantially shorter than in the similar structures and lacks the DXXT motif implied in the binding of uridine rich RNAs (*Muto & Yokoyama, 2012*). While loop-2 is located on the side of the RRM domain that usually does not participate in the RNA binding, the length of loops-3 and 5 might be functionally relevant.

## Comparison with other RRM structures in complex with RNA

Focusing on the top 100 structures defined by Dali server (*Holm & Rosenstrom, 2010*) as most similar to AcRRM (Fig. 5B), 31 were RRMs in complex with RNA. They can be

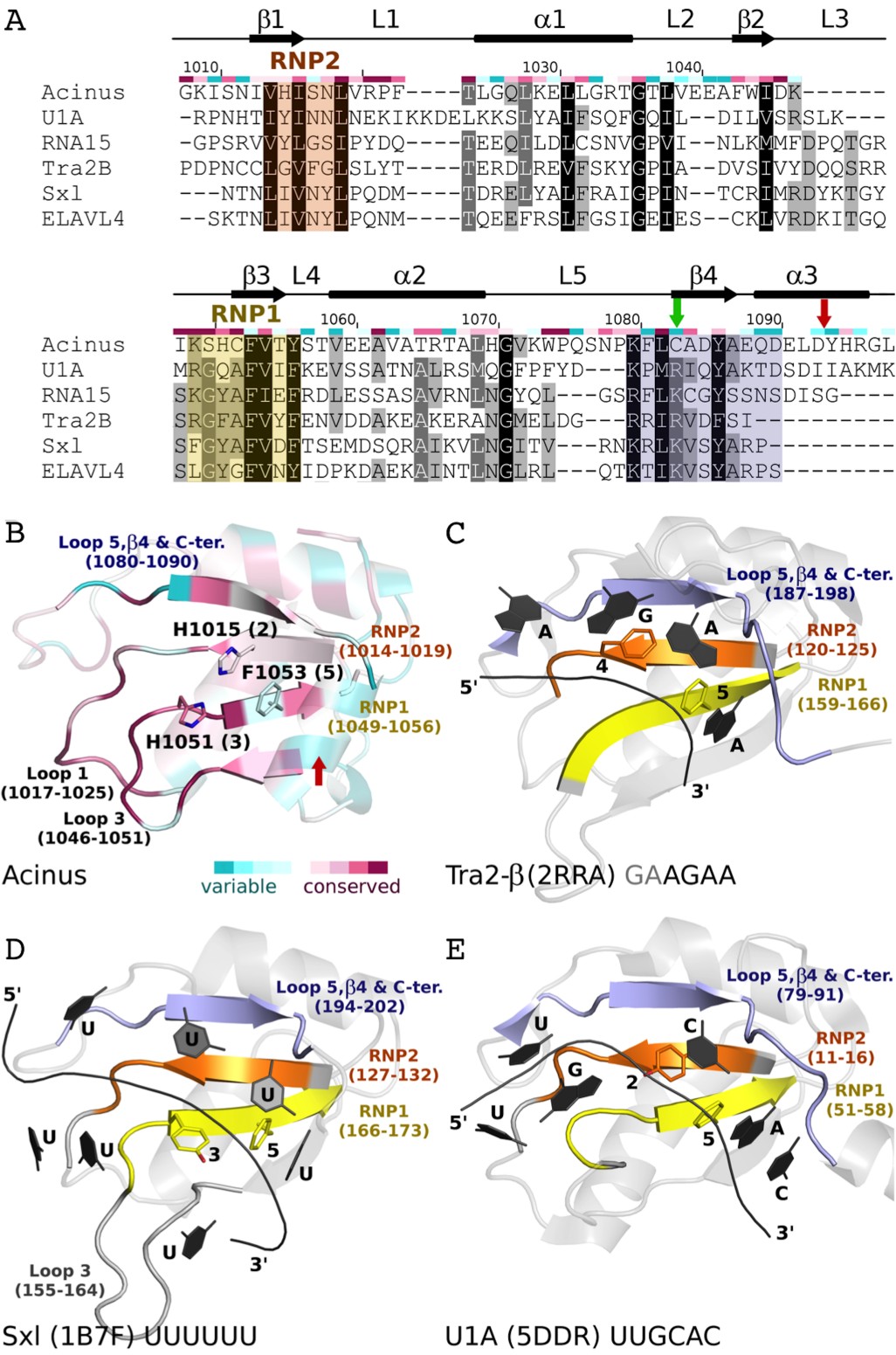

β1  L1  α1  L2  β2  L3

RNP2

|        | 1010 | 1030 | 1040 |
|--------|------|------|------|
| Acinus | GKISNIVHISNLVRPF----TLGQLKELLGRTGTLVEEAFWIDK------ |
| U1A | -RPNHTIYINNLNEKIKKDELKKSLYAIFSQFGQIL--DILVSRSLK--- |
| RNA15 | -GPSRVVYLGSIPYDQ----TEEQILDLCSNVGPVI--NLKMMFDPQTGR |
| Tra2B | PDPNCCLGVFGISLYT----TERDIREVFSKYGPIA--DVSIVYDQQSRR |
| Sxl | ---NTNLIVNYLPQDM----TDRELYALFRAIGPIN--TCRIMRDYKTGY |
| ELAVL4 | --SKTNLIVNYLPQNM----TQEEFRSLFGSIGEIES--CKLVRDKITGQ |

β3  L4  α2  L5  β4  α3

RNP1

|        | 1060 | 1070 | 1080 | 1090 |
|--------|------|------|------|------|
| Acinus | IKSHCFVTYSTVEEAVATRTALHGVKWPQSNPKFLCADYAEQDELDYHRGL |
| U1A | MRGQAFVIFKEVSSATNALRSMQGFPFYD---KPMRIQYAKTDSDIIAKMK |
| RNA15 | SKGYAFIEFRDLESSASAVRNLNGYQL---GSRFLKCGYSSNSDISG---- |
| Tra2B | SRGFAFVYFENVDDAKEAKERANGMELDG---RRIRVDFSI---------- |
| Sxl | SFGYAFVDFTSEMDSQRAIKVLNGITV---RNKRLKVSYARP--------- |
| ELAVL4 | SLGYGFVNYIDPKDAEKAINTLNGLRL---QTKTIKVSYARPS-------- |

Loop 5,β4 & C-ter. (1080-1090)

H1015 (2)

F1053 (5)

RNP2 (1014-1019)

RNP1 (1049-1056)

H1051 (3)

Loop 1 (1017-1025)

Loop 3 (1046-1051)

Acinus

variable    conserved

Loop 5,β4 & C-ter. (187-198)

RNP2 (120-125)

RNP1 (159-166)

Tra2-β(2RRA) GAAGAA

Loop 5,β4 & C-ter. (194-202)

RNP2 (127-132)

RNP1 (166-173)

Loop 3 (155-164)

Sxl (1B7F) UUUUUU

Loop 5,β4 & C-ter. (79-91)

RNP2 (11-16)

RNP1 (51-58)

U1A (5DDR) UUGCAC

**Figure 5 Comparison of AcRRM with structurally related RRM domains crystallized in complex with RNA.** (A) Sequence alignment of AcRRM and five structurally most similar RRM domains. RNP1 and RNP2 are marked with yellow and orange shading, respectively. The additional fragment, comprising β-strand 4 and flanking regions, that might provide sequence specific contacts with RNA is shown in purple. AcRRM secondary structure elements (loops: 1–5 (L1–5)) and sequence conservation based on the scores calculated with the ConSurf program (Fig. S4) (*Ashkenazy et al., 2010*) are given above the alignment. (B–E) Comparison of the putative RNA binding site of (B) AcRRM with the known RNA complexes of similar RRMs: (C) Tra2-β, (D) Sxl and (E) U1A. AcRRM is colored according to the ConSurf conservation scores, the other RRMs are colored in light gray apart from RNP2 (orange), RNP1 (yellow) and the additional fragment covering β4 (purple). The caspase cleavage site in AcRRM is indicated by red arrow, the cysteine residue forming a disulphide bond in the crystal by green arrow. In panels (C–E), only the RNA fragments interacting with RNP1 and RNP2 are shown and the numbers 1–5 correspond to the position in RNP1 and RNP2. The thin black lines indicate the RNA backbone. The complete picture of discussed RRM–RNA interactions is provided in Fig. S5.

divided into two major groups: RRMs that bind single-stranded RNA and RRMs that bind an RNA hairpin.

The first group includes, among others, RRMs of splicing factors: (i) Tra2-β in complex with the (GAA)$_2$ sequence (PDB code 2RRA, (*Tsuda et al., 2011*) Fig. 5C), where the AGAA tetra-nucleotide is recognized through the N- and C-terminal extensions, as well as stacking interactions on the β-sheet surface; (ii) Sex-lethal protein (Sxl) which binds to the 5′-UGUUUUUUU-3′ via a V-shaped, strongly electropositive cleft formed by β-sheets of two consecutive RRM domains (PDB code 1B7F, (*Handa et al., 1999*) Fig. 5D); (iii) HuD (ELAVL4) which binds (AU)-rich elements (AREs) in the 3′ untranslated regions of many short-lived mRNAs via RNPs and C-terminus (PDB codes 1G2E and 1FX1 (*Wang & Tanaka Hall, 2001*)). The nucleotide sequences of all these ligands align poorly in the analyzed structures, but in all of them the RNA is placed in proximity of RNP1 (contacts are made with U, G and A residues) and RNP2 (contacts are made with U, G and C residues).

The second group comprises multiple structures of U1A RRM in complex with the same RNA loop sequence (AUUGCACUCC) (Fig. 5E) and one structure of the Sxl RRM in complex with UUUGAGCACGUGA (PDB code 4QQB (*Hennig et al., 2014*)) (nucleotides that are in contact with RNPs are underlined).

## Putative RNA binding mode of AcRRM

RNA recognition motif domains are known for having diverse RNA binding modes (*Maris, Dominguez & Allain, 2005*; *Clery, Blatter & Allain, 2008*; *Muto & Yokoyama, 2012*; *Daubner, Clery & Allain, 2013*) that make the precise prediction of the Acinus interaction with the ligand very difficult. However, there are a few features that are likely to be conserved. In many RRMs, including the ones most similar to AcRRM, the RNA is bound at the face of the β-sheet with its 5′-end on the side of the β4 and 3′-end on the side of the β1-strand (Fig. 5). It seems likely that Acinus binds the RNA in analogous way. The RRMs usually use aromatic residues in position 2 of RNP2 (Fig. 5E) and 5 of RNP1 (Figs. 5C–5E) for the stacking interactions with the RNA bases. H1015 and F1053 are present in these positions in AcRRM and are likely to fulfil the same roles (Fig. 5B). The aromatic residue

in position 3 of RNP1 is often stacked against the ribose groups of the bound nucleic acid (Fig. 5D). This position is occupied by a H1051 in AcRRM that might have the same function (Fig. 5B). The first amino acid of RNP1 is often positively charged and sometimes interacts with the phosphate backbone. K1049 in AcRRM might play this role, but homologous residues of the similar RRMs are not involved in the interactions with the RNA backbone. While RNP1 and RNP2 mediated stacking interactions with the bases provide the RNA binding affinity, the sequence specificity determinants of RRMs are often located in surrounding secondary structure elements. The β4-strand and the flanking regions provide part of these interactions in RRMs similar to AcRRM (indicated with blue color in Fig. 5), but the protein-RNA hydrogen bonding patterns are too variable for precise predictions (Fig. S5).

Some RRM domains can accommodate a spectrum of RNA sequences, and the actual contribution of RNPs to ligand discrimination may not always be exclusive (*Clery, Blatter & Allain, 2008*; *Daubner, Clery & Allain, 2013*). The determinants of sequence specific RNA recognition may be located in other structural elements of RRMs. For instance, RRM2 of Sxl (*Handa et al., 1999*) and RRM1 of PABP (*Deo et al., 1999*) interact with RNA via loops, in addition to RNPs. In extreme cases, the RRM domains do not bind RNA through the β-sheet at all but they do so through other structural elements, for instance qRRMs use loops-1, 3 and 5 (*Dominguez et al., 2010*). The comparison of AcRRM with similar RRM structures suggests that analogous elements might play a role in RNA binding (Fig. 5). These include the above-mentioned β4 strand and its flanking regions (residues 1080–1090) that may provide sequence specific interactions. Alternatively, loop-1 (residues 1017–1025) and/or loop-3 (residues 1046–1051) that are immediately adjacent to the RNPs and characterized by high sequence conservation may be involved in RNA binding by AcRRM (Figs. 3B and 5B).

## DISCUSSION

Acinus has an RRM most similar to that of the RRM1 of Sxl from *Drosophila melanogaster* (PDB code: 4QQB). They share 29% sequence identity. Sxl is involved in splicing and translation regulation during sex development in the fly (*Samuels, Deshpande & Schedl, 1998*) and binds to polypyrimidine tracts through both of its RRM domains connected by a flexible linker. The linker does not participate in RNA binding, however, upon binding it becomes ordered as a short, distorted α-helix (*Crowder et al., 1999*; *Handa et al., 1999*). Structurally, AcRRM was found to be most similar to RRMs of several other factors implicated in RNA processing: Rna15, U1A, Tra2-β and HuD.

There are several known cases of RRM oligomerization *in vivo*. For instance, the RRM3 of HuR and the RRM of RBPMS homodimerize through a helical region and bind to RNAs through their β-sheets, located opposite to the dimerization site (*Scheiba et al., 2014*; *Teplova et al., 2016*). Despite the fact that AcRRM crystalized as a homodimer, in which two AcRRM monomers are bound via C1083 of their β4, we believe that any potential oligomerization of Acinus protein is unlikely to be performed by RRM domain in the reducing environment of the nucleus.

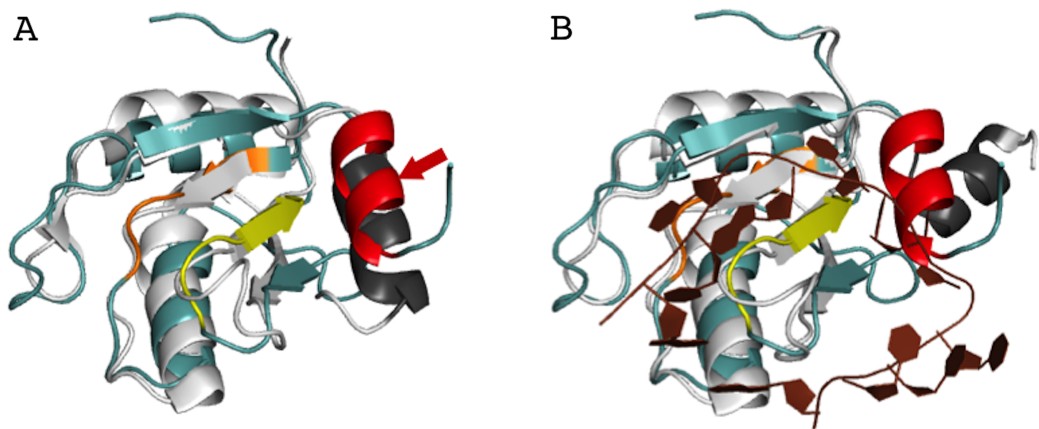

**Figure 6 Superposition of AcRRM with the RRM of U1A protein (PDB code 4C4W (*Huang & Lilley, 2014*)).** (A) Without RNA (B) in complex with RNA (dark brown color), with demonstration of the C-terminal α-helix relocation upon RNA binding. AcRRM is colored as in Fig. 2, and U1A in light gray with C-terminal α-helix highlighted in dark gray. The caspase cleavage site in AcRRM is indicated by red arrow (for better clarity only in panel A).     

In order to better understand the role of Acinus in splicing and apoptosis, more information is needed about its RNA binding mode. If AcRRM bound RNA in the way observed in related structures, the C-terminal α-helix would have to move away in order to make space for the RNA molecule. Such adjustment seems possible, and it is observed in some homologs with known structures. A good example is U1A RRM crystallized in complex with RNA (PDB code 4C4W (*Huang & Lilley, 2014*)). In this structure, the RRM domains that do not bind RNA have their C-terminal α-helix perfectly overlapping with the C-terminal α-helix of AcRRM (Fig. 6A). In the same crystal, in the RRM domains that bind RNA, the helix moves away to make space for the ligand (Fig. 6B). The C-terminal α-helix of U1A has been reported to cover the β-sheet surface and occlude the RNA binding (*Law et al., 2013*). Similar observations have been made for the N-terminal RNA-binding domain of human polyadenylation factor CstF-64 (*Perez Canadillas & Varani, 2003*). However, the C-terminal α-helix of CstF-64 does not relocate but unfolds upon RNA binding and extends into the hinge domain. Consecutively, this conformational change may facilitate the assembly of the polyadenylation complex (*Perez Canadillas & Varani, 2003*). The C-terminal α-helices are present also in the C-terminal RRM of La protein (*Jacks et al., 2003*) and the RRM of GW182 protein (*Eulalio et al., 2009*).

Interestingly, the 10-amino acid-long C-terminal α-helix of AcRRM has a caspase-3 target sequence and can be truncated by this protease down to six amino acids. It has been shown that Acinus is cleaved from both the N- and C-termini in order to produce the p17 form of the protein, which plays a role in apoptosis and chromatin condensation (*Sahara et al., 1999*). Due to the fact that the C-terminal α-helix contains the DELD caspase-3 target site and two acidic side chains, it has a highly electronegative potential (Fig. 3C). Therefore, it could shield the hydrophobic, electropositive surfaces in the absence of RNA (which could otherwise involve in non-sequence specific binding of the

ligand). The function of the C-terminal α-helix of AcRRM and the way in which the protease cleavage influences its conformation needs to be assessed by further biochemical and structural studies. For instance, it would be interesting to determine whether the C-terminal α-helix is (i) relocated upon RNA binding, destabilized and cleaved by caspase-3 (ii) destabilized and cleaved by caspase-3 ahead of RNA binding via RNPs or, alternatively, (iii) C-terminal α-helix occludes the RNA binding site on the β-sheet but RNA is bound through the loops?

## CONCLUSIONS

We crystallized and determined the structure of the RRM domain of the human Acinus protein spanning residues 1008–1100, at a resolution of 1.65 Å. The classical βαββαβ core of AcRRM is supplemented by an additional C-terminal α-helix packed against the β-sheet at the side opposite to the other two helices. The aromatic residues of the RNPs in AcRRM are exposed to the cavity potentially creating an RNA-binding area which is occluded by the C-terminal α-helix in the absence of the ligand. The additional α-helix can be found in some other RRM domains (e.g., of U1A or La protein) but it is generally considered to be an auxiliary structural element that may block the RNA-binding site and has to change conformation to enable ligand binding. Interestingly, this α-helix bears a caspase-3 target site and undergoes cleavage during apoptotic signaling. Further studies should address how caspase-3 cleavage changes the conformation of the AcRRM C-terminus and how it influences ligand binding.

## ACKNOWLEDGEMENTS

We thank Dr Justyna Czarnecka and Dr Radosław Pluta for the X-ray diffraction data collection. We thank Dr Agnieszka Kiliszek and Dr Matthew Merski for critical reading of the manuscript. We thank HZB for the allocation of synchrotron radiation beamtime and the assistance during data collection. HC thanks the tutors of the DLS-CCP4 Data Collection and Structure Solution Workshop and in particular Dr Ronan Keegan for the help with the structure solution.

### Funding

This work was supported by the Polish National Science Center grants: 2012/04/S/NZ1/00729 to Martyna Nowacka and 2015/17/B/NZ1/00861 to Humberto Fernandes. Humberto Fernandes was additionally supported by the Foundation for Polish Science grant HOMINGPLUS/2013-7/5. Honorata Czapinska was supported by the Polish National Science Center grant 2014/13/B/NZ1/03991 and 2014/14/M/NZ5/00558. Janusz M. Bujnicki was supported by the statutory funds of IIMCB. There was no additional external funding received for this study. The funders had no role in study design, data collection and analysis, decision to publish, or preparation of the manuscript.

## Grant Disclosures

The following grant information was disclosed by the authors:

Polish National Science Center grants: 2012/04/S/NZ1/00729 to Martyna Nowacka, 2015/17/B/NZ1/00861 to Humberto Fernandes, 2014/13/B/NZ1/03991 and 2014/14/M/NZ5/00558 to Matthias Bochtler.

Foundation for Polish Science grant HOMING PLUS/2013-7/5 to Humberto Fernandes.

## Competing Interests

The authors declare that they have no competing interests.

## Author Contributions

- Humberto Fernandes performed the experiments, analyzed the data, prepared figures and/or tables, authored or reviewed drafts of the paper, approved the final draft.
- Honorata Czapinska performed the experiments, analyzed the data, prepared figures and/or tables, authored or reviewed drafts of the paper, approved the final draft.
- Katarzyna Grudziaz performed the experiments, authored or reviewed drafts of the paper, approved the final draft.
- Janusz M. Bujnicki conceived and designed the experiments, analyzed the data, contributed reagents/materials/analysis tools, authored or reviewed drafts of the paper, approved the final draft.
- Martyna Nowacka conceived and designed the experiments, performed the experiments, analyzed the data, contributed reagents/materials/analysis tools, prepared figures and/or tables, authored or reviewed drafts of the paper, approved the final draft.

## Data Availability

The raw data is supplied as a Supplementary File and at the Protein Data Bank (ID 6g6s).

## Supplemental Information

Supplemental information for this article can be found online at http://dx.doi.org/10.7717/peerj.5163#supplemental-information.

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
