# Peer review of "Crystal structure of human Acinus RNA recognition motif domain"

_PeerJ, doi:10.7717/peerj.5163_

## Round 0.1 · original submission · Minor Revisions

As you can see, all three reviewers provided very positive and constructive comments. Please address all critiques raised by all reviewers and revise your manuscript accordingly.

Reviewer 1 ·

Basic reporting

The manuscript is well-written to present the structure of the Acinus RRM domain. There are a few points which could be improved:

Line 33: "includes the RRM domain" - This makes it sound as though the RRM is part of p17 and not vice versa.

Line 36: location of the caspase-3 target sequence - the text in parenthesis - typically, the references are made starting from the N-terminus and going towards the C-terminus. Also, the current description with 4 AA apart is a little non-scientific.

Line 46: Eponymous - please simplify to ensure international audience can easily understand the text

Lines 76-94: The introduction section gives a through background on Acinus and is extremely well-cited. But lines 76-94 come across as lines of facts stated, which are not particularly adding to the background. In my opinion, some of the references could be combined together to make it sound like a story. And, maybe even include some background on the C-terminal helix (Optional).

Line 182: Artifact spelling

Line 260: "right in the middle" - it is a little colloquial

Figure 2 Legend last line - it should be disulfide "bond"

I would like to add that the figures are well made and the consistency in the color scheme is definitely helpful to the reader. The color for the RNA strand in Figure 6B could be changed to better distinguish it (optional).

Experimental design

No comment.

Validity of the findings

Line 215-232: The section on "comparison with other RRM structures" - the manuscript presents a good comparison of Acinus RRM with other existing structures. While, the article discusses the findings from overlaying with RNA15 and U1A (basically figure 4 and 6), the discussion for figure 5 is missing. Rest of the figures have a clear finding that adds to the story and conclusions of the paper. But, what is being learnt from figure 5 that gets added to the conclusion of the entire paper?

I would like to add that the comparison with RNA15 and U1A are well presented.
Though for figure 4A : the PDB for RNA15 shows the presence of a C-terminal helix, which is missing in the figure.

Additional comments

I would like to commend the authors on successfully solving the structure of Acinus RRM. Overall, the manuscript is well written and conveys the relevance of the work.

·

Basic reporting

The article is written in a clear language and is easy to read.
The introduction is well written and serves to place the research into the structural and biology context. Nevertheless I found the sentence in line 71 a bit confusing because it is not clear to me if serine phosphorylation refers to caspase-3 residues or to acinus (they seem to be very far from the RRM). It is also unclear to me if caspase-3 also causes cleavage at N-terminus of Acinus p17 form
Figures are of high quality, although it is advisable to include labels for secondary structure elements for easy following descriptions in the text. In particular when comparing with other RRM structures.
References are well chosen in general. Perhaps the example of Csft64 (Perez-Canadillas and Varani 2003, EMBO J) should be mentioned in line 255 as an early case of a C-terminal helix blocking the RNA binding interface.
The PDB code (6G6S) of AcRRM should be reported in the article and in Figure 2

The PDB validation report has been provided. The structural data is of good quality.

Experimental design

Molecular biology and crystallographic methods are described in good detail.

I noticed this RRM domain has not been automatically annotated in the protein database, perhaps due to non-canonical RNP1 and RNP2 motifs. Because no experimental binding data has been presented, discussion about the RNA recognition properties should be based in a thorough bioinformatics analysis. However I believe that it is poorly presented in this case. For instance in figure 3 B a conservation map is shown over the structure, however it is not clear which Acinus homologues have been chosen to build this map. A figure with a sequence alignment of acinus homologues from different species should be provided at least as supplementary data.
I have check (blast) that Acinus RRM is very similar to Fox1 with sequence identity much higher (around 31%) than for UA1 and RNA15. Structural comparison with Fox1 should be included in the analysis (lines 200 to 208)

Validity of the findings

The article describes the structure of human Acinus RNA recognition motif that contain a basic RRM-like fold with an extra C-terminus helix. Although there are similar structures with this extra secondary element in similar configuration, the novelty of Acinus RRM is that the C-terminal helix contains the cleavage site of caspase-3, which is being proposed by the authors as a possible regulator of RNA binding activity. Nevertheless such hypothesis is only presented at theoretical level and there is not experimental validation. The article would benefit from doing simple binding experiments (e.g. EMSA with Acinus RRM and C-terminal truncated mutants).
The protein crystalizes as a disulphide crosslinked dimer (Figure 2). In my opinion, authors overlook the potential biological significance of this finding. They mention cysteine 1083 is conserved across species but at the same time claim that the disulphide bridge is probably a crystallization artefact. It should be noticed that the RNA binding properties of a dimer would be dramatically different to the monomer. Hence a more detailed investigation of the oligomerization state of this protein in solutions seems mandatory to me. I would suggest simple gel filtration experiments comparing wild type and C1083 mutant (e.g. to Ser) behaviour under oxidizing and reducing conditions.

In my opinion some parts of the article are of little relevance and describe vague concepts:
Lines between 209 and 213 describe two very small insertions in loops, which I think of little significance giving the high variability of loops across the different RRMs. I recommend removing this paragraph.
Lines 269-278 discuss some common themes in RRM RNA recognition, suggesting that Acinus RRM might follow similar strategies. But there are no real evidences/clues/hints in this work that support any of these mechanisms, apart that the RRM fold. I regard this part as superfluous

Additional comments

At the light of the structure, the potential biological significance of AcRRM dimerization should be experimentally addressed or at least discussed in comparison with other RRM homodimers

Comparative band shift experiments with wild-type and a C-terminal truncated version (at caspase site) should be performed to provide some experimental foundations for the discussion.

Reviewer 3 ·

Basic reporting

The manuscript is well written for the most part, but there are several grammatical errors which make sentences confusing, as well as several instances where words appear to be missing. The use of commas needs to be improved throughout the manuscript.

The figures are well put together, but there are some issues with clarity.

The following issues should be addressed with the manuscript and figures before publication...

1) Figure 4A and 4B. Change the colors of the other RRM’s to be different from one another. Having both be light grey and dark grey gives the impression that you are showing two different conformations of the same protein. Additionally, the left most black arrow in 4A is pointing to a different spot in 4B. My understanding is that they should both be pointing to the same place.
2) Figure 5C-5E. The light grey is very hard to see. Change this to a more visible color. Additionally, what is the purple highlighting representing in these figures? I did not see this stated in the caption for 5B-E, but it was mentioned for 5A. Is it the same thing for B-E? Please clarify.
3) Figure 5 Caption. Silver-Blue is not the best way to describe the color. Purple would be better.
4) Figure 5A. Why is RNP2 presented first, when throughout the manuscript and in the caption RNP1 is mentioned before RNP2? Move the RNP1 alignment to before the RNP2 alignment, unless this was for a specific reason.
5) Figure 2 Caption. Change “bound” to “bond” and change “artefact” to “artifact”. Also, clarify in the caption that the disulfide bond is between C1083 from different chains.
6) Capitalize the letters in all presented PDB codes (i.e. PDB code 5FJ4).
7) Line 124. Add “a” before HiTrap.
8) Change all mentioned amino acid residues to one letter code and number throughout the manuscript. On line 68, for example, the residue is listed as aspartic acid 1093, but D1093 is more appropriate.
9) In the Crystallization portion of the methods, please provide the vendor for the Index Screen. This is especially important as this screen gave the crystal used for data collection.
10) Line 171. Change “restrains” to “restraints”
11) Line 182. Change “artefact” to “artifact”
12) In the Results section, Figure 2A is never referred to.
13) Line 195. Change “extents” to “extends”
14) Line 212. Change “the loop 5” to “loop 5”
15) Line 242. Change “few” to “several”
16) Please revise the sentence starting at line 244 and ending at 246. It is a bit unclear in its current form.
17) Line 267. Point (ii) is very unclear. Please revise.
18) Line 63-64. Please add commas to this sentence. In its current form it seems to be a run-on sentence and is a bit unclear. Commas in the appropriate places would help the clarity.

Experimental design

No comment.

Validity of the findings

The structure reported is novel, and seems to be the first reported structure of the human Acinus RRM domain, but the authors never explicitly state this fact. A cursory search of the PDB returns no other published structures of this protein. If it is indeed the first time this protein structure is being reported, this fact should be added to the manuscript.

Additional comments

I would recommend this manuscript for publication, provided all issues are addressed.

---

## Round 0.2 · accepted · Accept

In the revised version, all the critical points raised by the reviewers were completely addressed, and the manuscript was amended accordingly. Therefore, the revised version is acceptable in present form.

#